# Spatial Distribution Pattern and Sampling Plans for Two Sympatric *Tomicus* Species Infesting *Pinus yunnanensis* during the Shoot-Feeding Phase

**DOI:** 10.3390/insects14010060

**Published:** 2023-01-09

**Authors:** Chengxu Wu, Siyu Chen, Maofa Yang, Zhen Zhang

**Affiliations:** 1College of Forestry, Guizhou University, Guiyang 550025, China; 2Guizhou Provincial Key Laboratory for Agriculture, Pest Management of the Mountainous Region, Institute of Entomology, Scientific Observing and Experimental Station of Crop Pest in Guiyang, College of Agriculture, Guizhou University, Guiyang 550025, China; 3Key Laboratory of Forest Protection of National Forestry and Grassland Administration, Research Institute of Forest Ecology, Environment and Protection, Chinese Academy of Forestry, Beijing 100091, China

**Keywords:** beetles, *Tomicus*, Taylor’s power law, Green model, field sampling, aggregation distribution

## Abstract

**Simple Summary:**

*Tomicus minor* (Hartig) and *Tomicus yunnanensis* Kirkendall and Faccoli are two sympatric species that infest *Pinus yunnanensis* (Franchet) in southwest China. Sound knowledge of the two pine shoot beetles’ distribution within *P. yunnanensis* is needed to formulate accurate sampling methods. Three pine forests with different experimental sites in Yuxi, Yunnan province, China were investigated from 2016 to 2018. Field counts were modeled in various spatial models, and it was determined that two species adults showed aggregated spatial distributions. A fixed-precision sampling plan showed that, for a *D* of 0.25 and 0.10, sample sizes of 41 plants and 259 plants for *T. minor* and 33 plants and 208 plants for *T. yunnanensis* were adequate, respectively. This sampling program could be useful for the integrated pest management of two sympatric *Tomicus* species.

**Abstract:**

*Tomicus minor* (Hartig) and *Tomicus yunnanensis* Kirkendall and Faccoli are two sympatric species that infest *Pinus yunnanensis* (Franchet) in southwest China, contributing to growth losses. Accurate sampling plans are needed to make informed control decisions for these species. We investigated three pine forests within experimental sites in Yuxi, Yunnan province, China from 2016 to 2018. The spatial distribution patterns of two pine shoot beetles during the shoot-feeding phase were determined using Taylor’s power law. The optimum sample sizes and stop lines for precision levels of 0.25 and 0.10 were calculated. The model was validated using an additional 15 and 17 independent field datasets ranging in density from 0.06 to 1.90 beetles per tree. *T. minor* and *T. yunnanensis* adults showed aggregated spatial distributions. For *T. minor*, sample sizes of 41 and 259 trees were adequate for a *D* of 0.25 and 0.10, respectively, while for *T. yunnanensis,* a mean density of one individual per tree required sample sizes of 33 plants (*D* = 0.25) and 208 plants (*D* = 0.10). The software simulations of this sampling plan showed precision levels close to the desired levels. At a fixed-precision level of 0.25, sampling is easily achievable. This sampling program is useful for the integrated pest management (IPM) of two sympatric *Tomicus* species.

## 1. Introduction

Eight pine shoot beetles of the genus *Tomicus* (Latreille) (Coleoptera: Curculionidae: Scolytinae) around the world [1,2] are destructive to conifer forests [3] and contribute to growth losses in the Palearctic [4,5,6,7,8]. Among them, *Tomicus minor* (Hartig) is widely distributed in Eurasia [9], while *Tomicus yunnanensis* Kirkendall and Faccoli has been re-identified as a new and highly aggressive species after molecular and morphological studies [1,10]. These two species have caused massive mortality to *Pinus yunnanensis* (Franchet) stands in southwestern China, where outbreaks are ongoing [11,12,13].

Adults of these two *Tomicus* species have the same maturation period [3]. Newly emerged adults fly to the crowns of nearby pine trees, usually in late spring, where they bore into shoots for the next seven to ten months, until they are sexually mature [14]. Although they only weakly attack the bole, these two pine shoot beetles aggregate densely during the shoot-feeding period in *P. yunnanensis,* subsequently reducing the resistance of the pine trees and facilitating the reproduction (mating and oviposition) of the beetles in the living trunks [14]. In recent studies, *T. yunnanensis* and *T. minor* were shown to often infest pine trees together [15,16], meanwhile, the attack pattern of the latter has evolved to be regulated by the former’s attack habits and time during the trunk-breeding phase [11]. Therefore, it is possible that interspecific competition affects the spatial distributions of sympatric *Tomicus* species. Interestingly, our previous study found that coexistence and homologous competition between *T. yunnanensis* and *T. minor* could be achieved through the allocation and compensation of spatial and temporal resources [17]. At the same time, the coexistence of groups of these two sympatric species with different population densities could have impacted prior semivariogram models and model parameters [18]. Interspecific competition between sympatric species may be an important factor regulating population distributions and dynamics [11,19]. After long-lasting and overlapping shoot-feeding periods, significant tree mortality occurs following bole attacks by *T. minor* and *T. yunnanensis*. The complexity of sympatric pests increases the difficulty of integrated pest management (IPM).

The development of IPM systems for insects requires an understanding of their spatial and temporal distributions for accurate population density estimation [20]; appropriate fixed-precision sampling plans are also basic components of a successful IPM program [21], as they improve the decision-making process by accurately estimating pest populations in the field [22]. Field sampling is often very time-consuming, and a 35–50% reduction in sampling effort can result from fixed-precision sequential sampling [23]. Determinations of the types and spatial distribution parameters are critical for the development of sequential sampling plans [24]. In the evaluations of the effectiveness of control measures, this method has been widely reported in the research on agricultural and forestry pests, such as *Scirtothrips dorsalis* Hood (Thysanoptera: Thripidae) in Florida blueberry [25], Citrus aphids (Hom., Aphididae) on two orange species [26], and *Mesoplatys ochroptera* Stål (Coleoptera: Chrysomelidae) on *Sesbania* [27]. However, there has been no effort to develop a fixed-precision sequential sampling plan for the sympatric *Tomicus* species infesting *P. yunnanensis*. The present study was designed to investigate the spatial distributions of *T. minor* and *T. yunnanensis* in *P. yunnanensis* forests and to develop sampling plans that will be useful for IPM.

## 2. Materials and Methods

### 2.1. Study Area

The study area was a nature reserve on Hongta mountain in Yuxi, Yunnan province, southwestern China, where two sympatric *Tomicus* species are perennially endangered in the same domain. The spatial distribution patterns were determined in October 2016, October 2017, and October 2018 in the reserve. The studies were conducted in three different artificial forest experimental sites of 20-year-old *P. yunnanensis* naturally infested by *T. minor* and *T. yunnanensis*. The experimental sites were as follows: EXP. A, Ketudi (24°18′34″ N, 102°34′33.34″ E), EXP. B, Guanyinshan (24°18′30″ N, 102°34′25″ E), and EXP. C, Tuoniaoyuan (24°18′27.89″ N, 102°34′27.75″ E). The average diameter of the survey shoots (diameter of the entrance hole) was 0.73 cm, and the average diameter at breast height of the Yunnan pine trees was 6.6 cm (range 2.1–10.1 cm) with an average tree height of 4.3 m (range 1.9–7 m). No pesticides were applied during the study period.

### 2.2. Field Sampling

The fields were sampled in October late in the shoot-feeding phase of the beetles, when the damage produced by the *Tomicus* species is obvious and easy to investigate. Three experimental sites were traced out, and counts were performed each year. Each experimental site was 10 rows in width (50 m) and 10 columns (50 m) in length, having an area of 2500 m^2^. The distance between the trees was approximately 5 m, of which GPS coordinates were recorded; in all, 100 trees in each sample plot were selected and inspected for beetles based on naturally yellow shoots, which were cut approximately 10 cm from the top of the tips with high-branch scissors because our investigation found that the entrance hole of *Tomicus* was approximately 4 cm to 7 cm away from the top of the tips. When the infection is not serious, there is only one beetle in a shoot. A sampling unit of one tree was collected. All possible damaged shoots from each sampled tree were cut, and the pine needles were removed; all shoots with beetles from each tree were placed in labeled 50-mL centrifuge tubes. Finally, all samples were brought back to the laboratory. The beetles were dissected from the shoots and identified based on their external morphological characteristics as described by Kirkendall et al. [1] and Li et al. [2] using a dissection microscope (Olympus SZX7, Olympus, Tokyo, Japan).

### 2.3. Statistical Analysis

Data were analyzed using SPSS 19.0 software (IBM Corp., Armonk, NY, USA). Each tree was considered a single replication for analysis. Taylor’s power law was used to evaluate the spatial distributions of the pine shoot beetle adults according to the following equation [28,29]:(1)$$\mathrm{lg}\left(S^{2}\right)=\mathrm{lg}\left(a\right)+b\times \mathrm{lg}\left(m\right)$$
where *S^2^* is the *Tomicus* population variance, *m* is the *Tomicus* population mean, coefficient *a* is the Y-intercept, and coefficient *b* is the slope of the regression line; lg(*a*) is the intercept, and *b* is the slope or the aggregation parameter. The distributions were considered aggregated (*b* > 1), random (*b* = 1), or uniform (*b* < 1). The general linear model regression procedure (GLM; SPSS) was used to compute the regression of the means and variances. Also, the regression coefficient (*R*^2^) was calculated to obtain goodness-of-fit to Taylor’s power law. The different linear regressions were tested for the equality of slopes by performing an analysis of covariance on the data obtained from different times of the season and during different years.

### 2.4. Fixed-Precision Sequential Sampling Plan

The optimum sample size (*n*) for estimating the *T. minor* and *T. yunnanensis* densities was calculated at two levels of fixed precision, 0.1 and 0.25, using the following equation [30]:(2)$$n=\frac{am^{\left(b-2\right)}}{D^{2}}$$
where *n* is the number of samples needed, *a* and *b* are the coefficients obtained from the regression of Taylor’s power law, *m* is the *Tomicus* population mean, and *D* is a fixed-precision level (0.10 or 0.25, which are acceptable for sampling for IPM purposes) [31].

Green’s formula [32] was used to establish the stop lines for the fixed-precision levels for sequential sampling of the *Tomicus* species,
(3)$$\ln\left(Tn\right)=\frac{\ln\left(D^{2}/a\right)}{b-2}+\frac{b-1}{b-2}\ln\left(n\right)$$
where *n* is the number of counted trees, *Tn* is the number of insects observed in *n* samples, *b* is the coefficient of Taylor’s power law, and *D* is the precision level (0.10 or 0.25).

### 2.5. Model Validation

To assess the reliability of Green’s sequential sampling plan, 15 and 17 additional independent datasets for *T. minor* and *T. yunnanensis* populations were collected in 2018 from 15 forests in Yuxi City. The beetles were sampled as described previously, but from 300–400 trees per plot. The validation was performed following the resampling approach, using the Resampling Validation of Sample Plans (RVSP) software developed by Naranjo and Hutchison [33] based on a resampling simulation technique. The simulations were conducted using 500 sampling bouts without replacement with a minimum sample size of 10 for a fixed-precision level of *D =* 0.25. At *D* = 0.10, resampling without replacement failed, and we tested this level of precision using resampling with replacement [33].

## 3. Results

### 3.1. Spatial Distribution of Two Sympatric Tomicus spp.

The mean number of *T. minor* individuals per plant was 0.08–0.43 (Table 1), while that of *T. yunnanensis* was 0.26–0.86. There were no significant differences between the slopes for the different years or for different species (different years: *df* = 8, *F* = 0.034, *p* = 0.859; different species: *df* = 8, *F* = 0.106, *p* = 0.754). Therefore, common regression was used to predict the lg(*S*^2^) versus lg(*m*) relationship. The slopes for *T. minor* and *T. yunnanensis* were significantly greater than 1.0 (*T. minor*: *b* = 1.444, *R*^2^ = 0.852, *t* = 6.352, *df* = 1, 7, *p* < 0.001; *T. yunnanensis*: *b* = 1.134, *R*^2^ = 0.621, *t* = 3.390, *df* = 1, 7, *p* = 0.012), The variances and means were significantly related as shown with Taylor’s power law, indicating an aggregated distribution of the two species in *P. yunnanensis* forests (Figure 1).

### 3.2. Fixed-Precision Sequential Sampling

The optimum sample sizes of *T. minor* and *T. yunnanensis* at the fixed-precision levels of 0.25 and 0.1 are presented in Figure 2. With increasing beetle density, the optimum sample size decreased rapidly. However, there was less variability among sampling bouts for any one field data set, particularly at mean densities greater than 2 insects per sample unit. Moreover, the optimum size at a fixed-precision level of 0.1 was always higher than that at a fixed-precision level of 0.25.

Numerical sample size curves were obtained from Taylor’s power law coefficients. A mean density of one *T. minor* individual per plant required a sample size of 41 plants for *D* = 0.25 and of 259 plants for *D* = 0.10. A mean density of one *T. yunnanensis* individual per plant required a sample size of 33 plants for *D* = 0.25 and 208 plants for *D* = 0.10 (Figure 2).

The calculated stop lines using Taylor’s model at two given precision levels are shown in Figure 3. The numbers of required shoot samples to cross the stop lines were significantly changed. The results indicated that the sampling of *T. minor* and *T. yunnanensis* must continue until the cumulative number of *Tomicus* spp. reached 24 and 28 beetles per tree at precision levels of 0.25 or 313 and 214 beetles per tree at precision levels of 0.10, respectively (Figure 3).

### 3.3. Validation of Developed Sampling Plan

The simulations performed with RVSP software for *T. minor* and *T. yunnanensis* populations produced average precisions (*D*) of 0.29 and 0.28, respectively; these were close to the desired precision of 0.25. On average, the sequential sampling model performed better than expected, and the precision was better than the pre-set value in only a few cases (Figure 4a,b). The mean optimum sample sizes for *T. minor* and *T. yunnanensis* were 150 and 77 trees at a precision level of 0.25 (Figure 4c,d), respectively.

## 4. Discussion

The spatial distribution models fit to the data showed a consistent aggregated distribution of the two *Tomicus* species in *P. yunnanensis* forests in Yuxi, Yunnan Province. A similar distribution pattern has been also observed for *T. piniperda* (L.) (Curculionidae, Scolytidae) [11,34] and *T. brevipilosus* (Eggers) (Curculionidae, Scolytidae) [35], which are closely related species to those in the present study [1]. This supported our previous research [17,18,36] in Pu’er city, where three *Tomicus* species compete and coexist. The distribution of the two species in the particular fields was stable between seasons, allowing the sampling plans to be used throughout the season in the *P. yunnanensis* plantations. The mean optimum sample sizes for *T. minor* and *T. yunnanensis* were 150 and 77 trees, respectively, at a precision level of 0.25.

Taylor et al. [37] indicated that the spatial distribution of a species is completely density dependent with a positive correlation between population aggregation intensity and population density. Although the densities of the two *Tomicus* species here were low, an aggregated spatial distribution of *T. minor* and *T. yunnanensis* was documented. The aggregated distributions of these two species in *P. yunnanensis* forests are likely due to their own aggregated behavior based on semiochemicals, which was confirmed in our previous studies [37]. Beetle concentrations in the shoots and subsequent feeding both considerably weaken the tree, decreasing its resistance to bole attacks and requiring a high number of stem attacks for the insects to overcome this resistance [38,39]. Other pine shoot beetles showed similar behaviors in pine plants, where adults moved slowly between plants and were aggregated in the trunks of certain pine trees [38]. Similar results were obtained for some *Tomicus* species, such as *T. piniperda* (L.) (Curculionidae, Scolytidae) [39,40,41], *T. brevipilosus* (Eggers) (Curculionidae, Scolytidae) [42], and *T. destruens* Woll. (Curculionidae, Scolytidae) [43]. Therefore, there is an opportunity to test field experiments using the push–pull strategy based on aggregation and anti-aggregation pheromones of *Tomicus* to develop potential IPM techniques. The present results were not consistent with those from other insects; for example, Shahbi and Rajabpour [44] showed that the spatial distribution of *Phthorimea operculella* Zeller (Lepidoptera: Gelechidae) tended toward randomness with low egg and larval densities.

The key limitation in this study was determining the effects of interspecific competition on the spatial distribution patterns. The results were consistent with a single population without interspecific competition [34], indicating that interspecific competition did not affect the spatial distribution patterns of these sympatric species. This could be the consequence of aggregation pheromones produced by *Tomicus* [45,46]; moreover, the *Tomicus* spp. were more strongly attracted to damaged shoots than to undamaged shoots, and they showed an attraction to shoots damaged by their own species [47]. The benefits of aggregation have been well established for ‘aggressive’ bark beetle species that primarily attack living trees. When adults can aggregate in pine trees, it subsequently reduces the resistance of the pine trees and facilitates the reproduction of the beetles in the living trunks [14,48].

The *Tomicus* densities and given precision levels are two key factors that determined the optimum sample sizes for the measurements of their populations. As a consequence of their aggregated distribution patterns, the numerical sample size for correct estimates of the *Tomicus* density can be very high at low population levels. Therefore, Green’s stop lines could be a suitable method for estimating the population density of these two sympatric *Tomicus* species. Meanwhile, the number of samples required to attain a certain precision seems to be a function of density; fewer samples are required at higher densities. This is the result of the relationship between the mean and the variance of the population density [49]. The average sample number in this case was 150 and 77 plants for *T. minor* and *T. yunnanensis*, respectively, which can be easily covered in 30 min by a single data collector (personal observation). For IPM purposes, this level of precision is acceptable [20].

When *Tomicus* individuals at high population levels attack the shoots, one shoot might be infected by multiple individuals, meanwhile [15], this will also affect our simulated result. However, the methods of spatial distribution according to the quantity of fallen shoots pruned by the pine shoot beetles and mass felling of the sample pine were difficult to carry out [50]. We then focused on the sequential sampling of hazards at different population densities. No previous studies have been performed to develop fixed-precision sequential sampling of the *Tomicus* species on Yunnan pine trees. Therefore, we cannot compare the present results with other studies. However, Naranjo and Hutchison [33] put forward that all fixed-precision sample plans normally perform poorly at low population densities (<0.2 insects per sample unit), which is similar to the density of *T. minor*. However, it is not clear for *T. yunnanensis* why the fixed-precision sample plans performed poorly at mean densities of 0.955–1.131. We recommend the use of Green’s model with a *D* of 0.25 for sampling of the *Tomicus* spp. However, achieving a precision level of 0.10 would require an average sample number of 313 and 214 trees for *T. minor* and *T. yunnanensis*, respectively, which is very time-consuming and not feasible for most IPM specialists.

## 5. Conclusions

No effects of interspecific competition on the populations’ spatial distribution patterns were found. The spatial distribution parameters of the two sympatric *Tomicus* species slightly differed with no significance. Further, the optimum sample sizes and fixed-precision sequential sampling stop lines were different for the two *Tomicus* species and desired precision levels. The sampling program developed here could be useful for IPM programs for these two sympatric *Tomicus* species.

## Figures and Tables

**Figure 1 insects-14-00060-f001:**
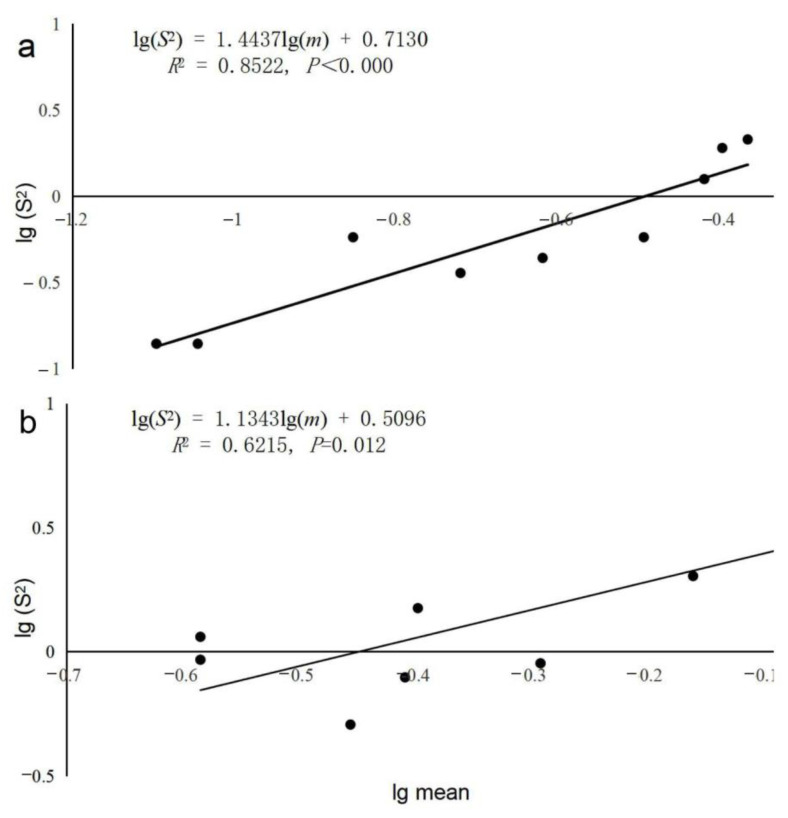
Linear relationship between the field variances and means of *Tomicus minor* (**a**) and *Tomicus yunnanensis* (**b**) infested *Pinus yunnanensis* collected at all experimental sites during the shoot-feeding phase with Taylor’s power law regression from 2016 to 2018.

**Figure 2 insects-14-00060-f002:**
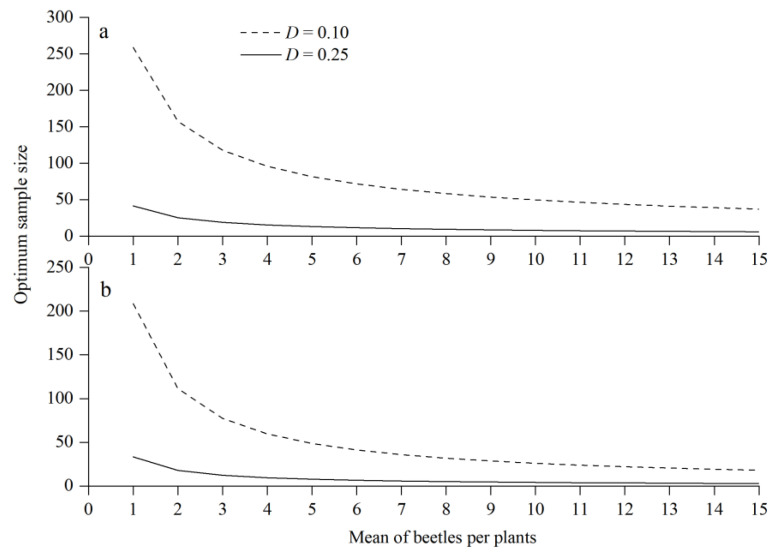
Optimum sample size of *Tomicus minor* (**a**) and *Tomicus yunnanensis* (**b**) in three *Pinus yunnanensis* tree sizes at precision levels of 0.1 and 0.25.

**Figure 3 insects-14-00060-f003:**
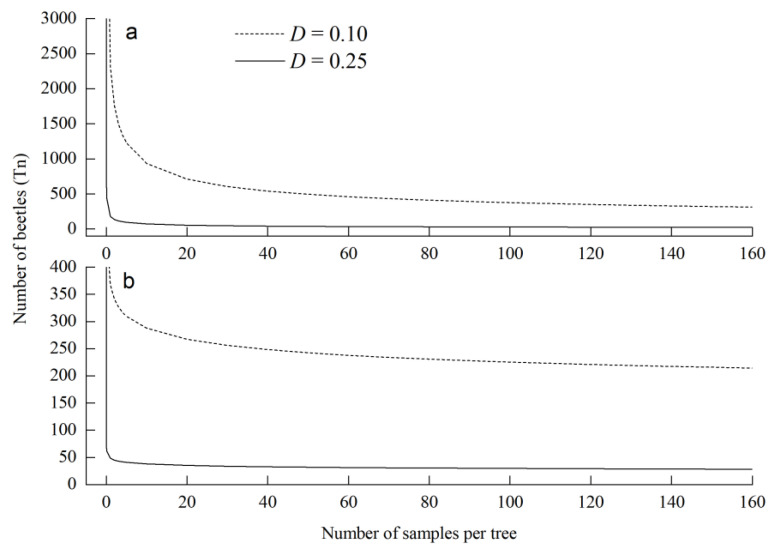
Green’s stop lines for fixed-precision sequential sampling of *Tomicus minor* (**a**) and *Tomicus yunnanensis* (**b**) in three *Pinus yunnanensis* tree sizes at precision levels of 0.1 and 0.25.

**Figure 4 insects-14-00060-f004:**
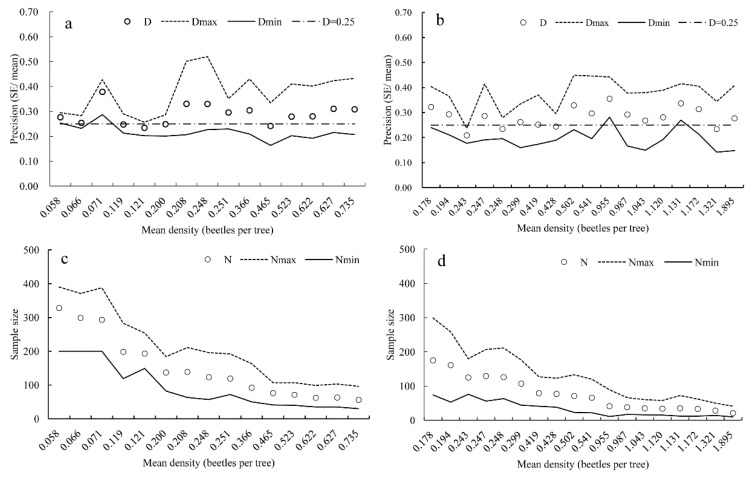
Resampling analyses of Green’s sequential sampling plan using independent field data for *Tomicus minor* (**a**,**c**) and *Tomicus yunnanensis* (**b**,**d**). The dash horizontal lines in (**a**,**b**) represent the desired level of precision (0.25).

**Table 1 insects-14-00060-t001:** Sampling of *Tomicus minor* and *Tomicus yunnanensis* infested on *Pinus yunnanensis* from different experimental sites during the late shoot-feeding phase from 2016 to 2018.

Sampling Data Sets	EXP	Yr	Num.	*Tomicus minor*			*Tomicus yunnanensis*		
				Mean	SE	Variance *S*^2^	Mean	SE	Variance *S*^2^
1	A	2016	100	0.38	0.11	1.26	0.51	0.09	0.90
2	B	2016	100	0.08	0.04	0.14	0.26	0.11	1.15
3	C	2016	100	0.40	0.14	1.91	0.26	0.10	0.93
4	A	2017	100	0.14	0.08	0.58	0.39	0.09	0.79
5	B	2017	100	0.09	0.04	0.14	0.40	0.12	1.50
6	C	2017	100	0.19	0.06	0.36	0.69	0.14	2.02
7	A	2018	100	0.32	0.08	0.58	0.86	0.19	3.46
8	B	2018	100	0.24	0.07	0.44	0.35	0.07	0.51
9	C	2018	100	0.43	0.15	2.14	0.85	0.19	3.54

EXP, experimental sites; Yr, year of sampling; Num., number of plants; SE, standard error.

## Data Availability

The data presented in this study are available in this manuscript.

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
