# Peer review of "Spatial Distribution Pattern and Sampling Plans for Two Sympatric Tomicus Species Infesting Pinus yunnanensis during the Shoot-Feeding Phase"

_insects, 2023, doi:10.3390/insects14010060_

Round 1
Reviewer 1 Report
This scientific manuscript summarizes a prodigious amount of work by the co-authors on an important topic. The paper reports the sampling program of important forest pests. It well describes the need in practice for an effective method reducing the amount of work needed in monitoring population levels of Tomicus minor and T. yunnanensis. Methodically the study is sound and the statistical approach is well carried out.
Comments:
General, that multiple attacks are the result of beetles competing for a limited number of suitable shoots. The number of attacks in individual shoots was not counted in the study. Therefore, please discuss this issue in the discussion on the basis of references.
Line 38–40: Tomicus is a genus of beetles belonging to the family Curculionidae and the Scolytinae subfamily.
I`d suggest of sentence: Eight pine shoot beetles of the genus Tomicus (Latreille) (Coleoptera: Curculionidae: Scolytinae) around the world [1-2] are destructive to conifer forests [Lieutier et al. 2015] and contribute to growth losses in Palearctic [Långström and Hellqvist 1990, 1991; Czokajlo et al., 1997; Borkowski 2001, 2006).
Lieutier F, Långström B, Faccoli M. 2015. The Genus Tomicus. In: Vega FE, Hofstetter RW (eds) Bark Beetles: Biology and Ecology of Native and Invasive Species. Elsevier Academic Press, Amsterdam, The Netherlands, pp 371–426.
Långström B, Hellqvist C. 1990. Spatial distribution of crown damage and growth losses caused by recurrent attacks of pine shoot beetles in pine stands surrounding a pulpmill in southern Sweden. J. Appl. Entomol. 110, 261–269.
Långström B, Hellqvist C. 1991. Shoot damage and growth losses following three years of Tomicus attack in Scots pine stand close to a timber storage site. Silva Fennica 25, 133–145.
Borkowski A. 2001. Threats to pine stands by the pine shoot beetles Tomicus piniperda (L.) and T. minor (Hart.) around a sawmill in southern Poland. J. Appl. Entomol. 125, 489–492.
Borkowski A. 2006. A spatial distribution of losses in growth of trees caused by feeding of pine shoot beetles Tomicus piniperda and T. minor (Col., Scolytidae) in Scots pine stands growing within the range of influence of a timber yard in southern Poland. J. For. Sci. 52, 130–135.
Czokajlo D, Wink RA, Warren JC, Teale SA. 1997. Growth reduction of Scots pine, Pinus sylvestris, caused by the larger pine shoot beetle, Tomicus piniperda (Coleoptera, Scolytidae), in New York state. Can. J. For. Res. 27, 1394–1397.
Line 40-42: I am sure that you should add extra data about Tomicus minor and Tomicus yunnanensis, something like this: Tomicus minor Hartig and Tomicus yunnanensis Kirkendall and Faccoli
Line 149-155: I`d suggest to write down Italic statistical symbols – for example: (different years: df = 8, F = 0.034, P = 0.859; different species: df = 8, F = 0.106, P = 0.754).
Table 1, explain the symbol abbreviations.
Figure 1. Please enter a P value.
References should be described according to Instructions for Authors (please correct the year).
Author Response
This scientific manuscript summarizes a prodigious amount of work by the co-authors on an important topic. The paper reports the sampling program of important forest pests. It well describes the need in practice for an effective method reducing the amount of work needed in monitoring population levels of Tomicus minor and T. yunnanensis. Methodically the study is sound and the statistical approach is well carried out.
Thanks you for your compliments on our experimental process and an important comment.
Comments:
General, that multiple attacks are the result of beetles competing for a limited number of suitable shoots. The number of attacks in individual shoots was not counted in the study. Therefore, please discuss this issue in the discussion on the basis of references.
These are important, valid points that we fully agree with. The required adjustments and changes have been made in the discussion on the basis of references. Please see L262-267.
Line 38–40: Tomicus is a genus of beetles belonging to the family Curculionidae and the Scolytinae subfamily.
I`d suggest of sentence: Eight pine shoot beetles of the genus Tomicus (Latreille) (Coleoptera: Curculionidae: Scolytinae) around the world [1-2] are destructive to conifer forests [Lieutier et al. 2015] and contribute to growth losses in Palearctic [Långström and Hellqvist 1990, 1991; Czokajlo et al., 1997; Borkowski 2001, 2006).
Lieutier F, Långström B, Faccoli M. 2015. The Genus Tomicus. In: Vega FE, Hofstetter RW (eds) Bark Beetles: Biology and Ecology of Native and Invasive Species. Elsevier Academic Press, Amsterdam, The Netherlands, pp 371–426.
Långström B, Hellqvist C. 1990. Spatial distribution of crown damage and growth losses caused by recurrent attacks of pine shoot beetles in pine stands surrounding a pulpmill in southern Sweden. J. Appl. Entomol. 110, 261–269.
Långström B, Hellqvist C. 1991. Shoot damage and growth losses following three years of Tomicus attack in Scots pine stand close to a timber storage site. Silva Fennica 25, 133–145.
Czokajlo D, Wink RA, Warren JC, Teale SA. 1997. Growth reduction of Scots pine, Pinus sylvestris, caused by the larger pine shoot beetle, Tomicus piniperda (Coleoptera, Scolytidae), in New York state. Can. J. For. Res. 27, 1394–1397.
Borkowski A. 2001. Threats to pine stands by the pine shoot beetles Tomicus piniperda (L.) and T. minor (Hart.) around a sawmill in southern Poland. J. Appl. Entomol. 125, 489–492.
Borkowski A. 2006. A spatial distribution of losses in growth of trees caused by feeding of pine shoot beetles Tomicus piniperda and T. minor (Col., Scolytidae) in Scots pine stands growing within the range of influence of a timber yard in southern Poland. J. For. Sci. 52, 130–135.
Thanks you for providing more valuable references.We have modified based on your valuable suggestions. Adding these references and revising the related sentences.
Line 40-42: I am sure that you should add extra data about Tomicus minor and Tomicus yunnanensis, something like this: Tomicus minor Hartig and Tomicus yunnanensis Kirkendall and Faccoli
Thanks you for your valuable advice. We have modified based on your valuable suggestions. Adding the namer of each species.
Line 149-155: I`d suggest to write down Italic statistical symbols – for example: (different years: df = 8, F = 0.034, P = 0.859; different species: df = 8, F = 0.106, P = 0.754).
We have write down Italic statistical symbols for symbols that need to be modified according your suggestion.
Table 1, explain the symbol abbreviations.
Figure 1. Please enter a P value.
Thank you for your valuable advice. We added P value. Please see Figure 1.
References should be described according to Instructions for Authors (please correct the year).
Thank you for comments. The required adjustments and changes have been made based on Instructions for Authors, especially correcting the year.
Reviewer 2 Report
Dear Authors,
In my opinion Your work meets the requirements to be published. However, it requires editorial and linguistic corrections (see corrections in the text).
But first of all, You should explain mire precisely what criteria determined to which category (slightly damaged, moderately damaged, heavily damaged) the plant material was applied. Such a division cannot be discretionary. Without this basic information, the results of the work are not comparable.
I also have doubts about the qualification of twigs of very different diameters (from 2.1 cm to 10.1 cm) for the experiment. These are branches of fundamentally different ages and with fundamentally different characteristics. It was unfortunate to group them in one pool for one experiment. I suggest carrying out a similar analysis in the future, but on groups of branches of similar diameter.
Best regards

Author Response
Dear Authors,
In my opinion Your work meets the requirements to be published. However, it requires editorial and linguistic corrections (see corrections in the text).
Thank you for your very careful advice. The required adjustments and changes have been made based on words provided by the reviewer. The name of each species has been added. The format of the reference has been modified.
But first of all, You should explain mire precisely what criteria determined to which category (slightly damaged, moderately damaged, heavily damaged) the plant material was applied. Such a division cannot be discretionary. Without this basic information, the results of the work are not comparable.
Thanks you for your compliments for methods. This description we did is a bit confusing, and the corresponding changes have been made. The degree of harm expressed here is relative in our original idea. However, strictly, the category (slightly damaged, moderately damaged, heavily damaged) of survey plots is based on Criteria for occurrence and disaster of forest pests in China (slightly damaged: 10%-20%; moderately damaged: 20%-50%; heavily damaged: > 50%). Please see the reference our previous research (Wu CX, Zang LP, Zhang SF, Kong XB, Liu F, Zhang Z, Li Y, Xu FL, Huang GY. Spatial distribution pattern of three sympatric Tomicus species initially infesting Pinus yunnanensis trunk (in chinese).Acta Ecologica Sinica, 2020, 40(11): 3646-365.). Obviously, our survey sample here does not meet this criteria. Therefore, we will not distinguish the degree of harm in our manuscript.
I also have doubts about the qualification of twigs of very different diameters (from 2.1 cm to 10.1 cm) for the experiment. These are branches of fundamentally different ages and with fundamentally different characteristics. It was unfortunate to group them in one pool for one experiment. I suggest carrying out a similar analysis in the future, but on groups of branches of similar diameter.
We are really grateful for professional and helpful comments. This is an very intelligent, valid comment that we fully agree with. This description we did is a bit confusing. We have advised the sentence. The average diameter of the survey shoots (Diameter of entrance hole) was 0.73 cm. Thank you again. Please see Lines 93-94.
Reviewer 3 Report
The manuscript “Spatial Distribution Pattern and Sampling Plans for Two 2 Sympatric Tomicus species infesting Pinus yunnanensis during 3 the shoot-feeding phase” examines the spatial dispersal patterns of 2 Tomicus spp. on Yunnan pine and developing an enumerative sequential sampling plan for making pest management decisions. There seems to be confusion about the appropriate use of the data sets that were collected with regard to the use of RVSP software. Ideally, data sets that represent the expected range of field conditions and insect densities should be collected over multiple locations and seasons. Once all data have been recorded, a subset of the data sets are held aside for use in validation of the sampling plan and the remaining data sets used for generating sampling plan parameters (in this case TPL) for use in the resampling software. It would be best to randomly select the validation data sets across all data sets to represent the widest range of conditions that may be encounter for a given insect and host plant. The authors took a slightly different approach and collected data sets specifically for validation use during a single season, which could be appropriate assuming that a wide range of densities and conditions are represented in the data.
It appears that the authors have used the data sets designated for Taylor’s Power Law (TPL) calculations, in the resampling software, to also generate sample size requirements and have presented those results in the abstract (line 29 and 30) and in Figures 2 and 3. These results are not valid as the TPL values from the data would also be used in the resampling software so they are not independent.
Another issue that must be addressed is to define the sample unit. In line 100-101, it appears that “shoot” is the sample unit but more detail is needed to define what “shoot” means e.g., 10-15 cm at the end of a branch. In addition, sample unit selection should be at random and the text describes shoots being selected that are “yellow”. If this coloration is indicative of insect injury or insect presence then it would bias the sample. So more detail is needed to describe the sample unit and how it was selected. Also additional detail is needed to describe why a managed area of forest was used to collect data. Is this the intended use of the sampling plan or is the sampling plan intended to be used in unmanaged areas? If the data collected to create the sampling plan is from managed areas, then the use of the sampling plan would only be appropriate in managed areas, unless further steps are taken to confirm the effectiveness of using the sampling plan in unmanaged areas.
L87-89 – Provide more detail describing the damage categories of “slightly”, “moderate”, and “severe”
L143 – after “500” add “sampling bouts”
Table 1 – consider changing column title “Sampling” to “Sampling Data Sets” and “EXP” to “Damage Level” (e.g., slightly, moderate, severe)
Figure 1 – move y-axis to the left side of graph by the axis titles
L165 – what data sets are used in Fig. 2 and 3? These figures should be based on data from the validation data sets. Not from those used in TPL calculations.
L175-176 – why are 3 different “tree sizes” stated here? Is this supposed to be 3 different “damage levels”?
Figure 3 – this is not a stop line graph. See: “Hodgson, E.W., E.C. Burkness, W.D. Hutchison, and D.W. Ragsdale. 2004. Enumerative and binomial sequential sampling plans for soybean aphid (Homoptera: Aphididae) in soybean. Journal of Economic Entomology 97(6): 2127-2136”, Figure 4 for an example. Stop line graphs would be generated using data from the resampling software using the validation data sets.
L178 – states sample size of 41 required and in Abstract (line 29) states 32 samples required.
L195 – states “only in a few cases was the achieved precision poorer than the pre-set value (Figure 4a,b)”. Precision improves as the value declines so this is not the case but the opposite is true where the precision was better in only a few cases than the pre-set value.
L249 – states “The average sample number in this case was 50 plants”, where is this number coming from, in line 211 authors state “sample sizes for T. minor and T. yunnanensis 210 were 150 and 77 trees, respectively”?
L264-265 – please clarify, authors state “The spatial distribution parameters of two sympatric Tomicus species dif-264 fered.” But in line 150-151 authors state “No significant differences were found between slopes for the different years or for different species”
Author Response
The manuscript “Spatial Distribution Pattern and Sampling Plans for Two Sympatric Tomicus species infesting Pinus yunnanensis during the shoot-feeding phase” examines the spatial dispersal patterns of Tomicus spp. on Yunnan pine and developing an enumerative sequential sampling plan for making pest management decisions. There seems to be confusion about the appropriate use of the data sets that were collected with regard to the use of RVSP software. Ideally, data sets that represent the expected range of field conditions and insect densities should be collected over multiple locations and seasons. Once all data have been recorded, a subset of the data sets are held aside for use in validation of the sampling plan and the remaining data sets used for generating sampling plan parameters (in this case TPL) for use in the resampling software. It would be best to randomly select the validation data sets across all data sets to represent the widest range of conditions that may be encounter for a given insect and host plant. The authors took a slightly different approach and collected data sets specifically for validation use during a single season, which could be appropriate assuming that a wide range of densities and conditions are represented in the data. It appears that the authors have used the data sets designated for Taylor’s Power Law (TPL) calculations, in the resampling software, to also generate sample size requirements and have presented those results in the abstract (line 29 and 30) and in Figures 2 and 3. These results are not valid as the TPL values from the data would also be used in the resampling software so they are not independent.
This is an important, valid point that we fully agree with. We will do more work in the future to verify and expand its use. Here we try to provide advice and strategies for a nature reserve on Hongta mountain in Yuxi City as best we can, so we use data from a single season (the lasts the longest damage period of Tomicus). We will want to refine our sampling with as much data as possible (over multiple locations and seasons) next step. We first take the basic sampling data, then the optimum sample size (n) for estimating T. minor and T. yunnanensis densities was calculated at two levels of fixed precision, 0.1 and 0.25 based on the data mentioned above; Green’s formula was used to establish stop lines for fixed precision levels for sequential sampling of Tomicus species, and lastly then model validation was performed using independent field data. Similar reference: Lessio F, Alma A . 2006. Spatial Distribution of Nymphs of Scaphoideus titanus (Homoptera: Cicadellidae) in Grapes, and Evaluation of Sequential Sampling Plans. Journal of Economic Entomology, 99(2):578-582.
Another issue that must be addressed is to define the sample unit. In line 100-101, it appears that “shoot” is the sample unit but more detail is needed to define what “shoot” means e.g., 10-15 cm at the end of a branch.
Thanks you for comments on our experimental approach. The required adjustments and changes have been made. 100 trees in each sample plot were selected and inspected for beetles based on naturally yellow shoots, which were cut about 10 cm from the top of the tips with high branch scissors. Because our investigation found that the entrance hole of Tomicus was about 4 cm to 7 cm away from the top of the tips.
In addition, sample unit selection should be at random and the text describes shoots being selected that are “yellow”. If this coloration is indicative of insect injury or insect presence then it would bias the sample. So more detail is needed to describe the sample unit and how it was selected.
Thanks you for comments. We have added the details.
Also additional detail is needed to describe why a managed area of forest was used to collect data. Is this the intended use of the sampling plan or is the sampling plan intended to be used in unmanaged areas? If the data collected to create the sampling plan is from managed areas, then the use of the sampling plan would only be appropriate in managed areas, unless further steps are taken to confirm the effectiveness of using the sampling plan in unmanaged areas.
Thanks you for your compliments on our experimental approach and an important comment.We are really grateful for professional and helpful comments. We have added the details. “To provide sampling plan support and the most economical prevention and control strategies at a nature reserve on Hongta mountain in Yuxi, Yunnan province, southwestern China, where two sympatric Tomicus species are perennially endangered in the same domain”. Here we try to provide advice and strategies for nature reserve on Hongta mountain of Yuxi City as best we can, so we use data from a single season (the lasts the longest damage period of Tomicus). We will want to refine our sampling with as much data as possible (over multiple locations and seasons) next step.
L87-89 – Provide more detail describing the damage categories of “slightly”, “moderate”, and “severe”
The degree of harm expressed here is relative concept. There is no strict implementation of national standards. It's misleading because we delete the statement about the level of harm
L143 – after “500” add “sampling bouts”
Table 1 – consider changing column title “Sampling” to “Sampling Data Sets” and “EXP” to “Damage Level” (e.g., slightly, moderate, severe)
Thank you for your very careful advice. The required adjustments and changes have been made based on comments provided by the reviewer.
Figure 1 – move y-axis to the left side of graph by the axis titles
Thank you for your comments. The required changes have been made.
L165 – what data sets are used in Fig. 2 and 3? These figures should be based on data from the validation data sets. Not from those used in TPL calculations.
Thank you for your experienced comment, giving us another way to analyze data sets. All data sets here were our field sampling in three different experimental plots from 2016 to 2018, validation data sets were not included. Based on our understanding, we first take the basic sampling data, then the optimum sample size (n) for estimating T. minor and T. yunnanensis densities was calculated at two levels of fixed precision, 0.1 and 0.25 based on the data mentioned above; Green’s formula was used to establish stop lines for fixed precision levels for sequential sampling of Tomicus species, and lastly then model validation was performed using independent field data. For example, see the reference: Lessio F, Alma A . 2006. Spatial Distribution of Nymphs of Scaphoideus titanus (Homoptera: Cicadellidae) in Grapes, and Evaluation of Sequential Sampling Plans. Journal of Economic Entomology, 99(2):578-582.
L175-176 – why are 3 different “tree sizes” stated here? Is this supposed to be 3 different “damage levels”?
We have deleted the refusing sentences.
Figure 3 – this is not a stop line graph. See: “Hodgson, E.W., E.C. Burkness, W.D. Hutchison, and D.W. Ragsdale. 2004. Enumerative and binomial sequential sampling plans for soybean aphid (Homoptera: Aphididae) in soybean. Journal of Economic Entomology 97(6): 2127-2136”, Figure 4 for an example. Stop line graphs would be generated using data from the resampling software using the validation data sets.
Thank you for your very rigorous opinion and giving us the reference.
Here we had presented sequential sampling stop lines for fixed-precision levels in our manuscripts. Some references presented a similar approach. For example:
(1) Lessio F, Alma A . 2006. Spatial Distribution of Nymphs of Scaphoideus titanus (Homoptera: Cicadellidae) in Grapes, and Evaluation of Sequential Sampling Plans. Journal of Economic Entomology, 99(2):578-582.
(2) Farzaneh Alizadeh Kafeshani, Ali Rajabpour, Sirous Aghajanzadeh, Esmaeil Gholamian, and Mohammad Farkhari. 2018. Spatial Distribution and Sampling Plans With Fixed Level of Precision for Citrus Aphids (Hom., Aphididae) on Two Orange Species. Journal of Economic Entomology, 111(2): 931-941.
L178 – states sample size of 41 required and in Abstract (line 29) states 32 samples required.
The required adjustments and changes have been made.
L195 – states “only in a few cases was the achieved precision poorer than the pre-set value (Figure 4a,b)”. Precision improves as the value declines so this is not the case but the opposite is true where the precision was better in only a few cases than the pre-set value.
Thank you for your comments. The required adjustments and changes have been made.
L249 – states “The average sample number in this case was 50 plants”, where is this number coming from, in line 211 authors state “sample sizes for T. minor and T. yunnanensis 210 were 150 and 77 trees, respectively”?
Thank you for your comments. The required adjustments and changes have been made.
L264-265 – please clarify, authors state “The spatial distribution parameters of two sympatric Tomicus species dif-264 fered.” But in line 150-151 authors state “No significant differences were found between slopes for the different years or for different species”.
Our expression is not rigorous enough, although there are differences, but not to a significant degree. We corrected the wrong statement.
Round 2
Reviewer 3 Report
Appears appropriate changes have been made to address reviewer concerns.